

# Moana Ocean Future Climate V1.0: High Resolution Marine Climate Futures For The New Zealand Region

Christopher J. Roach[1,3], Joao Marcos A. C. de Souza[1,4], Erik Behrens[2], and Stephen J. Stuart[2]

[1]MetOcean Solutions, a division of Meteorological Service of New Zealand, Raglan, New Zealand
[2]National Institute of Water and Atmospheric Research, Wellington, New Zealand
[3]Now at Institute for Marine and Antarctic Studies, University of Tasmania, Hobart, Australia
[4]Now at Cooperative Institute for Research in Environmental Sciences, NOAA Physical Sciences Laboratory, Boulder (CO), United States of America

**Correspondence:** Christopher J. Roach (christopher.roach@utas.edu.au)

**Abstract.** We have produced a series of 5 km resolution future climate dynamic downscalings for the ocean surrounding New Zealand covering CMIP6 reference conditions; SSP2-4.5 and SSP3-7.0 emissions trajectories. These downscalings combine the Moana Backbone 5 km resolution ROMS configuration with lateral boundary forcing from the 15 km resolution New Zealand Earth System Model (NZESM) and atmospheric forcing from the New Zealand Regional Climate Model 12 km atmospheric model.

We validated our reference period downscaling against the Moana Ocean Hindcast and find reasonable agreement to the west and north of New Zealand, but significant disagreement in the region of the Sub-Tropical Front to the east and southeast of the domain. This disagreement is consistent with known issues with the version of NZESM used as forcing in this study.

We see similar relative rates of increase in Ocean Heat Content in the upper ocean and mode waters all around New Zealand, but in the deeper ocean the rate of warming is stronger in the Tasman Sea and Antarctic Circumpolar Current than in the Sub Tropical Front East of New Zealand. We examine the occurrence of Marine Heat Waves (MHWs) and find that the use of a "fixed" baseline or one that takes into consideration a long-term warming based on the historical period results in important differences in the estimated number of days under MHWs for mid and end-of-the-century scenarios.

## 1  Introduction

New Zealand Aotearoa's marine domain is 21 times larger than its land mass and comprises almost 1.7% of the world's oceans. This large area drives vital economic activities, corresponding to about NZ$5B per annum from commercial fisheries and aquaculture alone (Dixon and McIndoe, 2022). In addition, it corresponds to roughly two-thirds of the total value of ecosystem services annually. According to estimates by MacDiarmid et al. (2013), this could correspond to US$357 billion worth of services each year. Therefore, understanding the impacts of our changing climate on the main physical drivers of our ocean domain is vital for building a resilient and sustainable future, and for understanding the interconnections between regional and global processes.





New Zealand lies at the confluence of the Tasman Sea, Pacific Ocean and Southern Ocean (Figure 1), all of which play key roles in controlling the regional ocean circulation. To the North a system of fronts, currents and mesoscale eddies (often collectively called the Tasman Front) carries waters from the East Australian Current across the Tasman (Oke et al., 2019) before feeding into the East Auckland Current (EAuC)(Chiswell et al., 2015). This current flows along the northern coastline of New Zealand southeast from Cape Reinga before turning east offshore of the Bay of Plenty. The EAuC in turn feeds, first, the North Cape and East Cape Eddies (ECE) and then the East Cape Current (ECC) which follows the east coast of the North Island south before turning east along the northern slope of the Chatham Rise. From here, part of the ECC flow continues offshore to join the southern rim of the South Pacific Gyre and part of the flow recirculates in the form of the Wairarapa Eddy (WE) (Chiswell et al., 2015).

To the south of New Zealand, the northern edge of the Sub-Tropical Front (STF) approaches the south-west coast of the South Island, with the current bifurcating and forming the northward flowing Westland Current (WLC) and the southward flowing Fiordland Current (FLC) (e.g. Matear et al., 2013; Chiswell et al., 2015). Further east the STF turns north and crosses the western margin of the Campbell Plateau (Smith et al., 2013; Behrens et al., 2021a), the inshore component forming the Southland Current (SLC) (Sutton, 2003), before turning east on the southern flank of the Chatham Rise. Further south the Sub-Antarctic Front, the northern fringe of the Antarctic Circumpolar Current, flows first east and then northeast along the southern flank of the Campbell Plateau.

At intermediate depths (∼1000 m) to the west of New Zealand, a comparatively young and fresh (S<34.4) subtype of Antarctic Intermediate Water (AAIW) forms within the Sub-Antarctic Front (SAF); some of this AAIW moves north into the eastern Tasman Sea, but most is entrained into the SAF and flows eastward (Chiswell et al., 2015). The SAF waters move along the flank of the Campbell Plateau, before splitting just west of the Bounty Plateau, some of the waters move into the Bounty Trough and then along the southern flank of the Chatham Rise, while the rest turns south east. Further north older, saltier and low-oxygen AAIW subtypes originating in the Pacific gyre follow the Continental Slope from the Northland Peninsula and Bay of Plenty, around East Cape and then south before turning East along the northern flank of the Chatham Rise. At the East of the Chatham Rise these salty AAIWs mix into the SAF waters (Chiswell et al., 2015) before moving South East and following the northern flank of the SAF out into the Southern Ocean.

In the present study, we have two major goals: 1. To improve the understanding of how these ocean current systems will respond to atmospheric and ocean warming as a key element in estimating the impacts of climate change on the New Zealand marine environment. The behaviour of the ocean current systems around New Zealand is sensitive to mesoscale, sub-mesoscale and high-frequency dynamics such as eddies, tides and transient responses to wind.

2. Focusing on the ecosystem impacts, to understand the increase in ocean temperatures and the frequency and intensity of extreme events.

In particular for marine heatwaves, Kerry et al. (2022) showed how important ocean advection can be for the set up of events. This means that boundary currents and shelf circulation must be resolved for the impacts on coastal regions to be properly estimated.



However, the majority of climate models from the Coupled Model Intercomparison Project phase 6 (CMIP6) employ ocean model resolutions coarser than 100 km. For instance, of the 39 CMIP6 models considered in Lyu et al. (2020), only five have an oceanic resolution of 50 km or less. CMIP6 atmospheric resolutions are also relatively coarse, for instance Lin and Yu (2022) found of the 54 CMIP6 models they considered only 19 had an atmospheric resolution of less than 110 km. Coarse
atmospheric resolution means these models are unable to resolve small-scale winds which are important in the coastal domain. Additionally, most CMIP6 future climate experiments archive ocean variables at monthly resolution, meaning higher frequency sub-mesoscale variability and tides are poorly resolved, reducing the utility of CMIP6 products for examining factors such as larval connectivity between fisheries. Thus, our attention turns to regional climate downscaling.

Existing regional climate downscalings for the New Zealand region include the New Zealand Earth System Model (NZESM),
New Zealand Regional Climate Model (NZRCM) and the Moana Ocean Hindcast (MOH). NZESM (Williams et al., 2016; Behrens et al., 2020) is a coupled ocean-atmosphere model offering approximately 15 km in-ocean (eddy-permitting) and 130 km in-atmosphere spatial resolution for both historical and future CMIP6 scenarios. The available ocean data is archived at 5-day intervals, limiting NZESM's utility for near-shore Lagrangian connectivity studies and analysis of extreme events. NZESM is supplemented by the New Zealand Regional Climate Model (NZRCM) atmospheric downscaling which uses the Met Office
GA7.0/GL7.0 model (Walters et al., 2019) to downscale the NZESM atmosphere to 12 km resolution. MOH (de Souza et al., 2022) is an ocean model implemented using the Regional Ocean Model System (ROMS) version 3.9 and forced with historical oceanic and atmospheric reanalysis data. MOH includes the main forcing mechanisms important for coastal circulation (e.g. tides, inverse barometer effect, rivers, etc), has a 5 km spatial resolution, and output temporal resolution of 1 hour. The MOH configuration is also used as an operational forecasting tool.

We combined the MOH configuration with boundary and surface forcing from NZESM and NZRCM to produce a new product - the Moana Ocean Future Climate downscaling (MOFC). In this paper we validate a 16 year period (1994-2010) of MOFC forced with historical NZESM against data from MOH and then explore regional scale medium-term (2030-2060) and long-term (2070-2099) marine climate change under the CMIP6 SSP2-4.5 and SSP3-7.0 emissions trajectories.

The paper is laid out as follows: Datasets used for forcing or validation are described in Section 2; our model configuration
and climate change scenarios are discussed in Section 3; validation of MOFC against MOH is presented in Section 4; Results are presented in Section 5 and our conclusions are summarized in Section 6.

## 2 Data

### 2.1 Moana Ocean Hindcast

The Moana Ocean Hindcast (de Souza et al., 2022; de Souza, 2022) uses the ROMS v3.9 (Shchepetkin and McWilliams,
2005) numerical modelling software package. The MOH domain covers the region 161-185 °E and 52-31 °S with a grid of 467x397 cells, giving a horizontal resolution of about 5 km x 5 km, and runs from 1993 to 2020. The vertical grid is made up of 50 vertical layers on an s-coordinate system using the stretching function given in de Souza et al. (2015). Bathymetry was generated by merging the General Bathymetric Chart of the Oceans (GEBCO) with local data sources. Atmospheric forcing



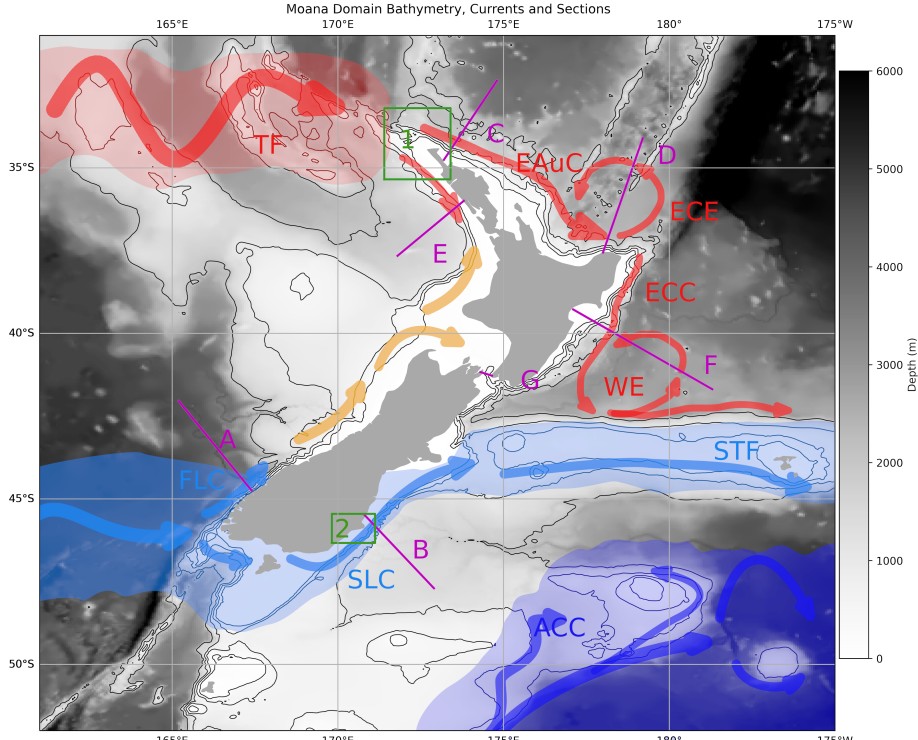

**Figure 1.** Bathymetry of the Moana model domain and major upper ocean current systems. The red shaded area indicates the Tasman Front (TF) and red arrows indicate currents drawing on the TF including the East Auckland Current (EAuC); East Cape Current (ECC); East Cape Eddy (ECE) and Wairarapa Eddy (WE). Orange arrows indicate currents dominated by waters from the Tasman Sea south of the TF. Light blue shading indicates the region of the Sub-Tropical Front (STF) and associated currents including the Southland Current (SLC) and Fiordland Current (FLC). Dark blue shading indicates the northern fringes of the Antarctic Circumpolar Current (ACC). Purple lines indicate the section on which the Transports given in Table 2 were computed. The green boxes indicate the region used to assess the sensitivity of Marine Heat Waves to detrending (1. Cape Reinga, 2. Otago Peninsula).

was taken from the Climate Forecast System Reanalysis (CFSR); lateral boundary forcing was taken from the Mercator Ocean
Global Reanalysis (GLORYS) 12v1 (Jean-Michel et al., 2021) and tidal forcing was generated from the global TXPO tidal solution (Egbert and Erofeeva, 2002).

    MOH has been extensively validated against observational data (de Souza et al., 2022; Kerry et al., 2022). MOH displays an SSH Root Mean Square Error (RMSE) of 0.11m and SST RMSE of 0.23°C. Relative to *in situ* profiles, MOH shows a potential temperature RMSE of 0.5-1°C in the upper ocean and less than 0.5°C below 500 m, and a salinity RMSE of 0.1-0.2g/kg in the
upper 500m and less than 0.1g/kg in the deep ocean. MOH was also evaluated against tide gauge data and coastal temperature observations, with good performance on both metrics.





## 2.2 New Zealand Earth System Model

NZESM (Williams et al., 2016) is a global coupled ocean-atmosphere model derived from the United Kingdom Earth System
Model (UKESM) (Kuhlbrodt et al., 2018). The ocean physics component uses the NEMO ocean engine (Madec, 2008; Gurvan
et al., 2022) on a 1° eORCA1 grid (approximately 100 km resolution) globally with two-way nesting to a 0.2° (12-20 km)
resolution grid in the region around New Zealand covering 132.7°E to 143.7°W and 60.17°S to 10.75°S. Atmospheric physics
is implemented with the Met Office Unified Model (Walters et al., 2019) on an N96 grid with an effective resolution of
about 130km at mid-latitudes. Sea-ice processes are modelled with CICE (Hunke et al., 2017); marine bio-geochemistry uses
MEDUSA (Yool et al., 2013) and land surface processes are modelled with JULES (Walters et al., 2019).

A total of twelve experiments were run with NZESM: future Shared Socioeconomic Pathway (SSP) 1 2.6, SSP2 4.5 and
SSP3 7.0 trajectories; a three-member historical baseline ensemble (1950-2014), and an extension of that ensemble through to
the end of the 21st century. The future climate scenarios cover a range of trajectories from little mitigation (SSP3 7.0) to heavy
mitigation measures (SSP1 2.6).

NZESM and its parent model, UKESM, have an equilibrium climate sensitivity of 5.3°C (Behrens et al., 2022; Meehl et al.,
2020), compared to a likely range of 1.5-4.5°C (Nijsse et al., 2020). These models have a transient climate response of 2.8°C
(Behrens et al., 2022; Meehl et al., 2020) compared to the likely range of 1-2.5°C (Nijsse et al., 2020). This implies the warming
seen in NZESM future climate simulations and our derived products are likely towards the higher end of warming under each
of the selected Shared Socioeconomic Pathways.

The performance of NZESM has been assessed by Behrens et al. (2020), who found, compared to the UKESM, NZESM's
enhanced regional resolution significantly improved representation of temperature (eliminating a 1-2°C cold bias) and transport
in the Tasman Sea (including the East Australian Current and the Tasman Front). NZESM also reduced the fresh bias in salinity
from about 0.7PSU in UKESM to about 0.5PSU.

During the development of the Moana Future Climate downscaling we identified a long-term sea surface height drift in
NZESM. Discussion with the NZESM development team indicates this drift is due to a previously undetected global imbalance
in the fresh water budget (E. Behrens, personal communication, 2021). As there is no observable spatial gradient in the bias,
this drift is unlikely to significantly impact the regional dynamics.

## 2.3 New Zealand Regional Climate Model

The New Zealand Regional Climate Model (NZRCM) is a limited-area atmospheric model based on the GA7 configuration
(Walters et al., 2019) and version 10.3 of the UK Met Office Unified Model (UM). The atmospheric model is coupled to a
land surface model provided by the Joint UK Land Environment Simulator (JULES) (Best et al., 2011). NZRCM runs over
a domain which is approximately 2600 km long and wide and spans New Zealand and the surrounding ocean, with 210x220
horizontal grid points at 0.11° (~12 km) resolution, a rotated coordinate north pole at 172 °E 50 °N, and 63 vertical levels in
the atmosphere.





NZRCM was forced by NZESM during the historical period (1950-2014) and the future period (2015–2100) under SSPs
2-4.5 and 3-7.0, via 6-hourly atmospheric lateral boundary conditions, monthly ozone and 5-daily sea surface temperatures,
while aerosols were prescribed as a monthly climatology. NZRCM forced MOFC via hourly mean surface fields of wind,
downward shortwave and longwave radiation, air temperature, relative humidity, precipitation and sea level pressure.

## 3   Method

### 3.1   Moana ROMS Configuration

In this study we employ the same Regional Ocean Modelling System (Shchepetkin and McWilliams, 2005) configuration
as MOH. However, we have migrated to ROMS4.0 to take advantage of the increased performance made available by the
parallelization of input and output routines. Otherwise, our configuration (Roach, 2024a) differs from MOH in the use of a
360 day calendar (as dictated by the forcing data) and the use of NZESM and NZRCM forcing in a one-way nesting setup.
Atmospheric forcing was primarily drawn from NZRCM with the exception of a narrow band of 1.9° latitude at the south of
the domain where NZESM was used instead; sensitivity experiments demonstrated that this approach did not induce significant
bias. Lateral boundary forcing was taken from NZESM and tidal forcing, as with MOH, is sourced from TXPO (Egbert and
Erofeeva, 2002).

To assess the effects of climate change on the ocean around New Zealand we conduct a total of five experiments consisting
of:

145       – A historical emissions run between 1990 and 2010

       – A high emissions (SSP3 7.0) medium term run from 2030 to 2060

       – A high emissions (SSP3 7.0) long-term run from 2070 to 2100

       – A medium emissions (SSP2 4.5) long-term run from 2070 to 2100

We did not conduct a medium term, medium emissions scenario as, on the global scale, SSP3 7.0 and SSP2 4.5 show little
divergence in radiative forcing and global mean temperature until the mid 2040s (Gidden et al., 2019).

## 4   Model Evaluation

To evaluate the performance of the present climate downscaling simulations we compared the run for the reference period
(MOFC) with a well-validated hindcast on which the configuration was based (MOH). By doing this, we can identify possible
biases that can contaminate the analysis of the climate change scenarios.



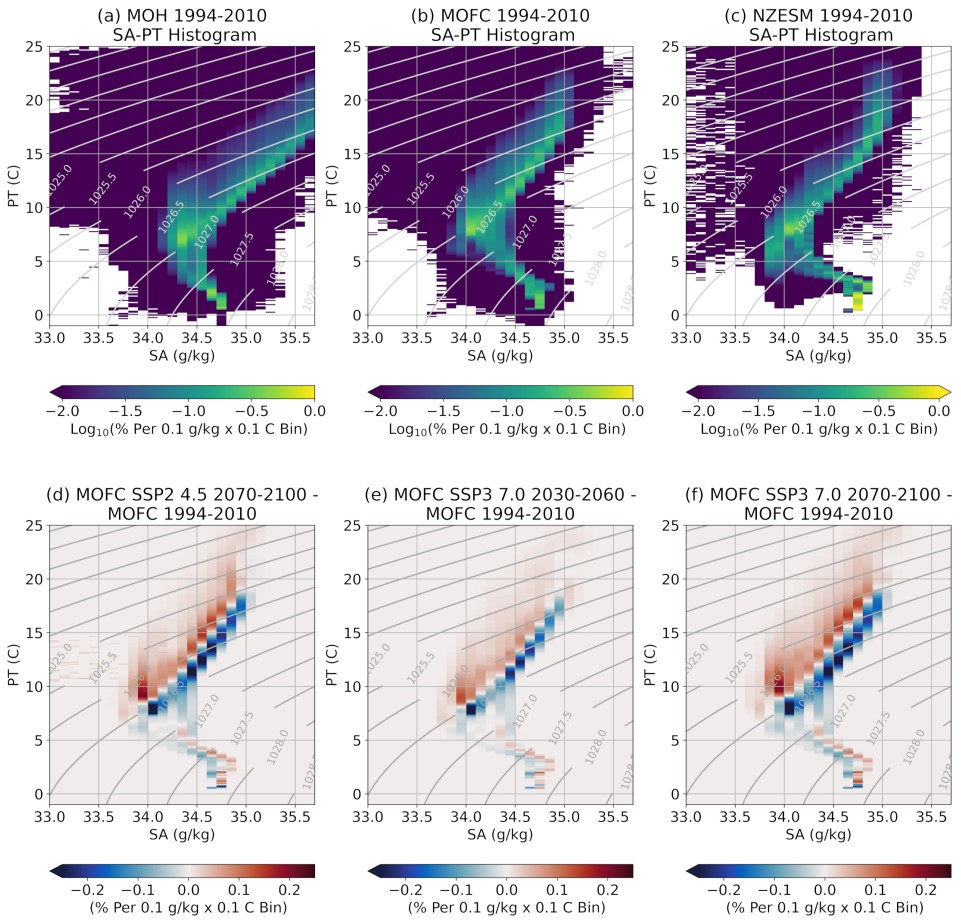

**Figure 2.** Potential temperature and absolute salinity diagrams over the entire domain for MOH 1994-2010 (a), MOFC 1994-2010 (b), NZESM 1994-2010 (c). Changes in potential temperature and absolute salinity diagrams relative to (b) for MOFC SSP2 4.5 2070-2100 (d), MOFC SSP3 7.0 2030-2060 (e) and MOFC SSP3 7.0 2070-2100 (f).

## 4.1 Temperature, Salinity and Sea Surface Height

Figure 2 shows temperature and salinity diagrams for the MOH, NZESM, and all of our experiments. MOFC has a fresh bias relative to MOH of about 0.2PSU at temperatures between 2°C and 12°C. At lower temperatures the salinity bias becomes smaller, while at higher temperatures the salinity bias increases to about 0.5 PSU (at 18°C). This salinity bias is consistent with the bias seen between MOH and NZESM (Figure 2c), implying it is likely due to the boundary and initial conditions derived from NZESM. We identify no systemic bias in potential temperature.

Maps of the mean and standard deviation of the sea surface temperature and the sea surface height are shown in Figure 3. The mean SSH maps (Figure 3a and b, domain-wide trend removed before averaging) show general consistency in large-scale structures between MOH and MOFC. However, there are several differences in detail: MOFC shows a region of slight negative





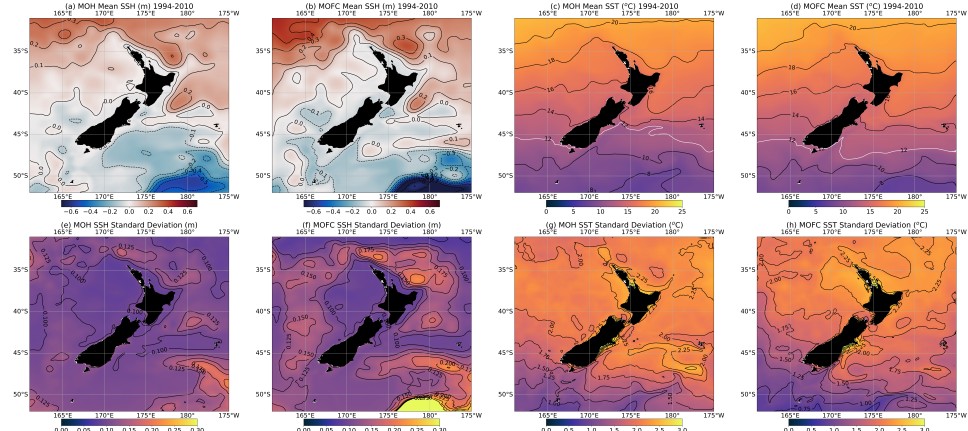

**Figure 3.** Sea Surface Height and Sea Surface Temperature averaged between 1994 and 2010 for (a and c) MOH and (b and d) MOFC. Standard deviations of SSH and SST between 1994 and 2010 for (e and g) MOH and (f and h) MOFC. The grey contours in c and d indicate the 12C isotherm.

SSH extending up the west coast of the North Island which is not present in MOH; MOFC positions the East Cape Eddy further
North-West than the MOH; MOFC shows an additional area of high SSH extending from the eastern boundary towards New
Zealand at about 47 °S, and MOFC shows a stationary eddy-like structure near the western end of the Bounty Trough ( 47 °S,
172-174 °E). Regions of enhanced variability, here measured by the standard deviation (Figure 3e and f), generally agree in
spatial structure, however, MOFC shows variability about 50% larger than MOH.

Mean SST (Figure 3c and d) shows good agreement in spatial structure and variability to the West, North-East and far South
of the domain. However, there is low agreement in the STF east of New Zealand and along the East Coast of the South Island
in proximity to the SLC, where MOH shows the 12 °C isotherm extending north to the Banks Peninsula, while MOFC shows
the 12 °C isotherm ending further south near the Otago Peninsula. Errors in the placement of the STF in the Tasman Sea and
East of New Zealand, and by implication errors in the SLC, have been observed in prior studies using eddy-resolving models
(e.g. Matear et al., 2013). Results in Behrens et al. (2020) indicate that, compared to observations, NZESM shows differences
in STF Sea Surface Height structure in the Bounty Trough and on the Southern slope of the Chatham Rise (their Figure 2) and
a warm bias in the STF East of New Zealand (their Figure 8). Evidence indicates that the error in the placement of the STF is
a feature of the version of NZESM used in this study and may reduce when using a higher resolution (4 km) two-way nested
model in the Tasman Sea-New Zealand region (Behrens et al., 2021b) or when the lateral boundary conditions are switched
to free slip in the nested region (Behrens per. coms.). The Mean Average Error between MOH and MOFC time-mean fields is
0.61 °C, reducing to 0.44 °C when the STF region off the East Coast is masked out. This compares to the Mean Average Errors
between MOH and NZESM which are 0.61°C and 0.50°C respectively.

To assess time-varying model performance we examined Empirical Orthogonal Functions (EOFs) of SST (Figure 4). Here
we show EOFs of sub-annual variability isolated using a high-pass Butterworth filter with a 1-year cut-off period. Since MOH




**Table 1.** Ocean Heat Content Trends (J per decade) for MOH, MOFC and NZESM during the historical period.

| Section | MOH 1994-2020 | MOFC 1991-2010 | NZESM 1991-2014 |
|---|---|---|---|
| 0-200 m | $7.35 \times 10^{20}$ | $7.88 \times 10^{20}$ | $8.01 \times 10^{20}$ |
| 200-1000 m | $1.80 \times 10^{21}$ | $-6.41 \times 10^{19}$ | $1.53 \times 10^{20}$ |
| 1000-2000 m | $4.90 \times 10^{20}$ | $2.28 \times 10^{20}$ | $4.46 \times 10^{20}$ |
| >2000 m | $3.25 \times 10^{20}$ | $6.95 \times 10^{19}$ | $-9.87 \times 10^{19}$ |

is forced with reanalysis data while MOFC is forced with data from a free-running Earth System model, variability in the forcing fields may not align in time and space (e.g. NZESM may not produce an El-Nino in the same year as the real world and MOH). Hence, a direct comparison of the time series of EOF principal components is not useful and we instead opt to compare the power spectra of the principal components (rightmost panels in Figure 4).

The first EOFs of SST (explaining 86% and 82.9% of variability, respectively) correspond predominantly to the seasonal cycle and show good agreement between MOH and MOFC in both spatial patterns and principal component power spectra. The second EOFs (explaining 1.8% and 2.4% of variability) also show good agreement in structure and PC spectra. The 3rd EOFs (explaining 1.0% and 1.3% of variability) and 4th EOFs (both explaining 0.5% of variability) show reasonable agreement in the large-scale spatial patterns and good agreement in principal component power spectra, but show disagreement in finer spatial scale features, with the third MOFC mode producing a larger region of negative values off the east coast of the South Island. We conducted a similar analysis for super-annual frequencies and sea surface height (not shown), with a similar agreement between the models.

These results show a good agreement for sea surface properties between MOH and MOFC, except for the region of the STF to the east of New Zealand and the SLC which feeds the STF. As discussed previously, the STF has been an area of difficulty in prior eddy-resolving model studies of the Tasman Sea - New Zealand region Matear et al. (2013). Examination of NZESM temperature and velocity sections (not shown) near the boundary of the MOH model domain evidence significant discrepancies in the STF region.

## 4.2 Ocean Heat Content

Time series of the domain-wide ocean heat content (OHC) are shown in Figure 5 for the upper ocean (panel a, 0-200 m), 200-1000 m (panel b), 1000-2000 m (panel c) and the deep ocean (panel d, 2000 m to seafloor). We find that in the upper ocean mean MOFC heat content is about 4% higher than MOH, but seasonal variability (indicated by the shaded areas) overlaps. Between 200 m and 1000 m, there is little seasonal variability and MOFC has about an 8% higher OHC than MOH. Between 1000 m and 2000 m MOFC OHC is approximately 3% higher than MOH, while in the deep ocean MOFC OHC is about 5% higher than MOH. These offsets are likely due to the relatively high climate sensitivity seen in NZESM and its parent model UKESM (Behrens et al., 2022; Meehl et al., 2020). Linear trends in domain-wide OHC are given in Table 1. Surface ocean (0-200 m) warming is in good agreement across all three models. In the 200-1000 m range, we see strong warming in MOH but little warming in either MOFC or NZESM. Between 1000 m and 2000 m, the MOFC shows about half as much warming as MOH or NZESM, and below 2000 m MOH displays significantly stronger warming than either MOFC or NZESM.



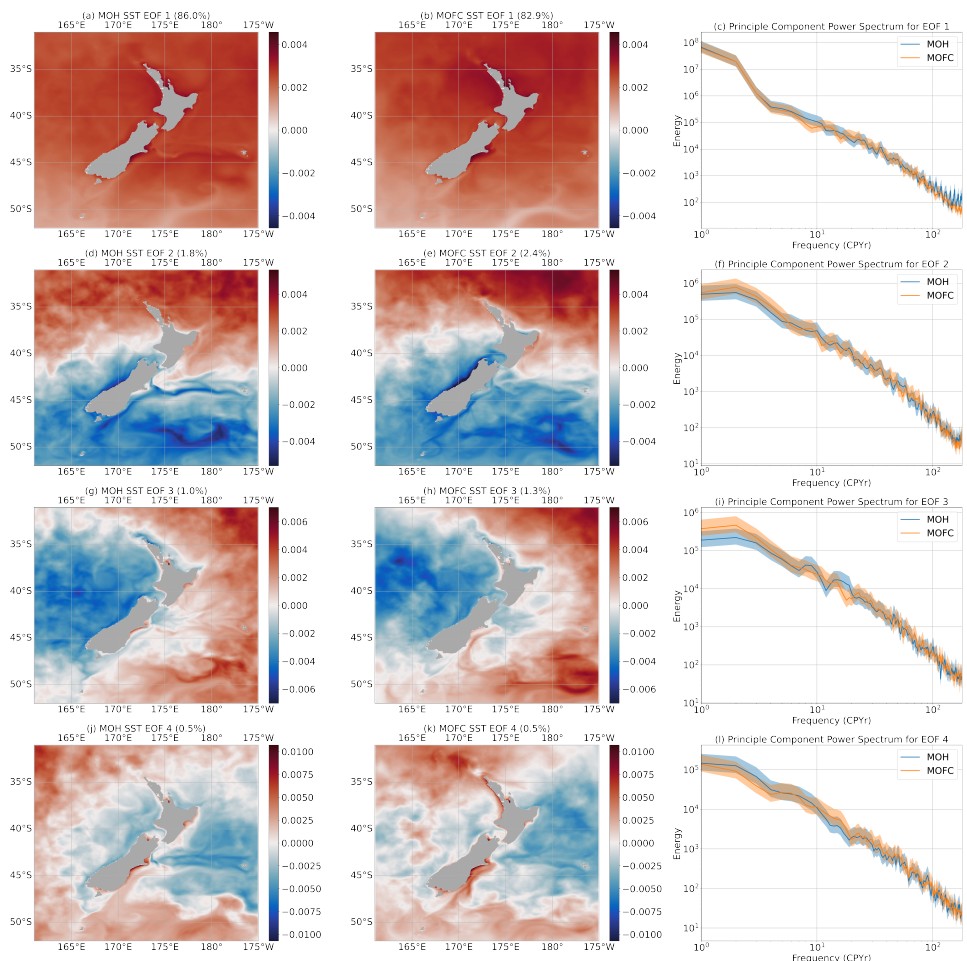

**Figure 4.** The first four EOFs of Sea Surface Temperature for the Moana Ocean Hindcast (left column) and Moana Ocean Future Climate (middle column). Power-spectra of principle components are shown in the right column, where MOH is shown in blue and MOFC in orange with 95% confidence intervals shown by the shaded areas.



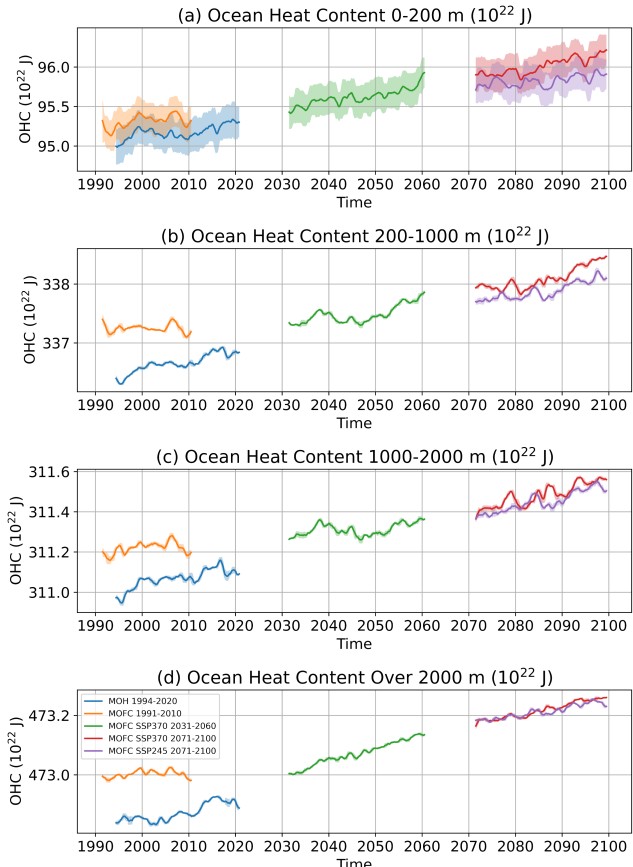

**Figure 5.** Time-series of Ocean Heat Content for the upper 200 m (a), 200-1000 m (b), 1000-2000 m (c) and 2000 m to the sea floor (d). Solid lines show the low-frequency component isolated using a 1 year rolling mean. Shaded areas indicate the standard deviation estimated in the same 1 year window.

Thus in the surface ocean MOH, MOFC and NZESM agree closely in OHC change. But at a deeper level, MOFC and NZESM generally produce slower warming than MOH. Examining maps of OHC trend by depth range (not shown) shows that the difference in warming between MOFC and MOH is primarily a product of the NZESM boundary forcing.

### 4.3 Transport and Vertical Structure

Time-mean vertical sections of temperature, salinity and cross-section velocity were computed for the lines shown in Figure 1. Example temperature and velocity sections for the EAuC, SLC and FLC are shown in Figure 6. The EAuC and FLC sections show good agreement in the time-mean temperature structure between MOH and MOFC. The velocity sections demonstrate that, in the EAuC and FLC, MOFC captures the coastal jets that form the core of currents and features in the deep ocean but shows some disagreement on the upper ocean structure offshore. The SLC, as previously seen in SST and SSH, shows





**Table 2.** Mean Transport (Sv) and Standard Deviation on sections shown in Figure 1 obtained from MOH, MOFC and other studies. Depths and distance ranges integrated are shown in the last column, FD indicates full depth of section and FW full width of section.

| Section | MOH (Sv) 1994-2010 | MOFC (Sv) 1994-2010 | Other Studies (Sv) | Max. Dist. (km) Max Depth (m) |
|---|---|---|---|---|
| (A) Fiordland Current | -4.18±2.82 | -6.19±4.07 | - | 50/FD |
| (B) Southland Current | 7.85±3.03 | 2.28±2.07 | 7.2±0.8, 10.6±1.0 Fernandez et al. (2018) 8.3±2.7 Sutton (2003) 12.4±4.5 Fernandez et al. (2018) | 150/FD |
| (C) EAuC (North) | 15.00±6.46 | 17.22±8.97 | 17.5±5.3 Stanton and Sutton (2003) | 220/1000 |
| (D) EAuC (South) | 12.97±11.21 | 12.35±12.35 | 14.8±3.2 Fernandez et al. (2018) 10-20 Chiswell and Roemmich (1998) | 200/FD |
| (E) ECC | -31.74±7.63 | -32.10±10.84 | >15 Chiswell (2005) | 250/FD |
| (F) West Coast of North Island | 4.45±2.19 | 5.07±2.63 | - | 250/FD |
| (G) Cook Strait | 0.28±0.23 | 0.40±0.31 | 0.25 Stevens (2014) 0.42±0.08 Hadfield and Stevens (2021) | FW/FD |

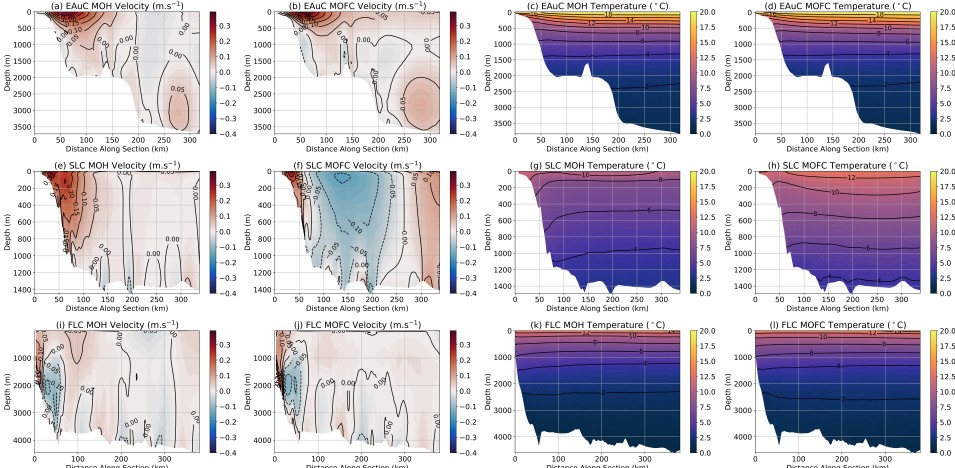

**Figure 6.** Mean Potential Temperature and Cross-Section Velocity (positive to North or East) sections for the East Auckland Current (a-d); Southland Current (e-h) and Fiordland Current (i-l). MOH sections are shown in the left and centre-right columns, and MOFC sections in the centre-left and right columns.)

significant disagreement between datasets. MOFC shows a warm bias throughout the water column and a large region of deep-reaching southward flow between 50 and 200 km offshore where the MOH reproduces the expected weak northerly flow.

We computed transport time series for each section, using a similar method to Kerry et al. (2022) in which we define time-evolving masks to isolate the cores of the currents. These masks were defined by a combination of a velocity magnitude >0.05 ms$^{-1}$ of the appropriate sign and a maximum depth and distance offshore as specified in Table 2. Once the core of the current was isolated, the velocities outside the core were set to zero, and trapezoidal integration was applied first vertically and then horizontally.

Average transport values and corresponding standard deviations are given for MOH, MOFC and selected prior studies in Table 2. As above, we find that MOFC shows good agreement with the MOH and observations, with the exception of the SLC.





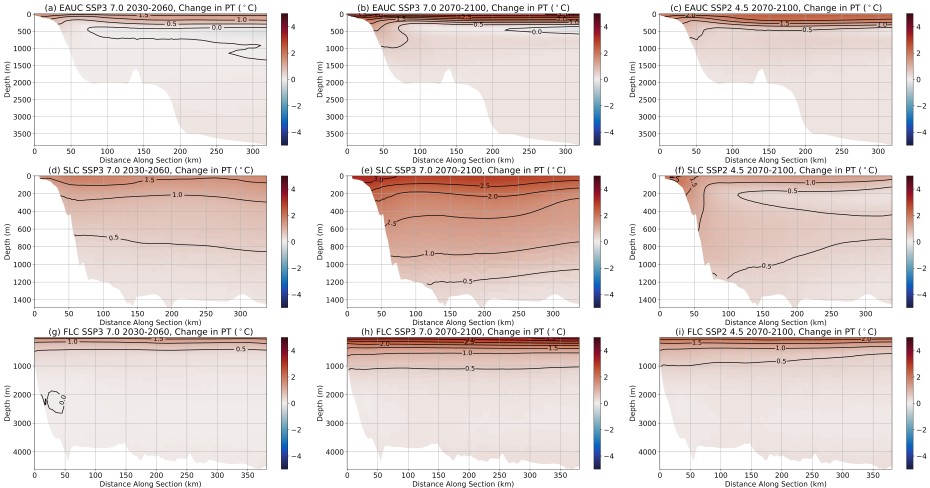

**Figure 7.** Temperature change on the East Auckland Current (a-c); Southland Current (d-f) and Fiordland Current sections (g-i) for SSP3 7.0 2030-2060 (left), SSP3 7.0 2070-2100 (middle) and SSP2 4.5 2070-2100 (right).

## 5  Future Climate Projections

### 5.1  Changes to Temperature, Salinity and Ocean Heat Content

Temperature-Salinity histograms for the future climate experiments are shown in Figure 2 (panels d, e and f). We see a shift towards warmer and fresher conditions predominately within the Subantarctic Mode Waters and Subtropical Mode Waters. Comparing the reference period (Figure 2b) and SSP3 7.0 late-century (Figure 2f), the inflexion point on the TS curve moves from 34.1 g/kg to 33.9 g/kg salinity, and temperature for the inflection point shifts from 7°C to 10°C. The high end of the TS curve, corresponding to warm Subtropical Waters in the north of the model domain, is warmer by about 2°C with little change in salinity. The other future climate experiments show similar but smaller changes. We see no significant changes in the lowest segment of the T-S curves, corresponding to intermediate and deep waters. Sections across the EAuC, SLC and FLC showing changes in temperature between the last decade of each experiment and the 1991-2010 reference period are shown in Figure 7. The EAuC (Figure 7a-c) shows substantial warming (>0.5°C) in the upper 500 m under all experiments. SSP3 7.0 2030-2060 shows slight cooling from 70km offshore at about 500-1000m depth; moving into both late-century experiments, this area of cooling shrinks, reducing in vertical extent and its inshore limit moving further offshore. In both late-century experiments, we also see a tongue of warmer waters following the slope down to between the 700 m and 1000 m isobaths. The SSP2 4.5 experiment does not show any offshore cooling, and as a result the ΔT=0.5°C contour reaches the 500 m level. Unlike SSP3 7.0 where the same contour only reaches the 350-400 m level. The SLC shows a generally coherent picture of warming for both SSP3 7.0 experiments with warming predominantly at the surface and spreading deeper into the water column in the late-century experiment. However, SSP2 4.5 shows warming much more confined to the surface and inshore, with a region of weak warming (ΔT<0.5°C) extending from 200-400 m between 120 km offshore and the eastern limit of the section.



**Table 3.** Ocean Heat Content Trends (J per decade) for the three future climate experiments.

| Depth Range | SSP 3 7.0 2030-2060 | SSP3 7.0 2070-2100 | SSP2 4.5 2070-2100 |
|---|---|---|---|
| 0-200 m | $1.0 \times 10^{21}$ | $1.12 \times 10^{21}$ | $5.04 \times 10^{20}$ |
| 200-1000 m | $1.31 \times 10^{21}$ | $1.97 \times 10^{21}$ | $1.57 \times 10^{21}$ |
| 1000-2000 m | $1.62 \times 10^{20}$ | $6.01 \times 10^{20}$ | $5.67 \times 10^{20}$ |
| >2000 m | $4.62 \times 10^{20}$ | $3.24 \times 10^{20}$ | $2.51 \times 10^{20}$ |

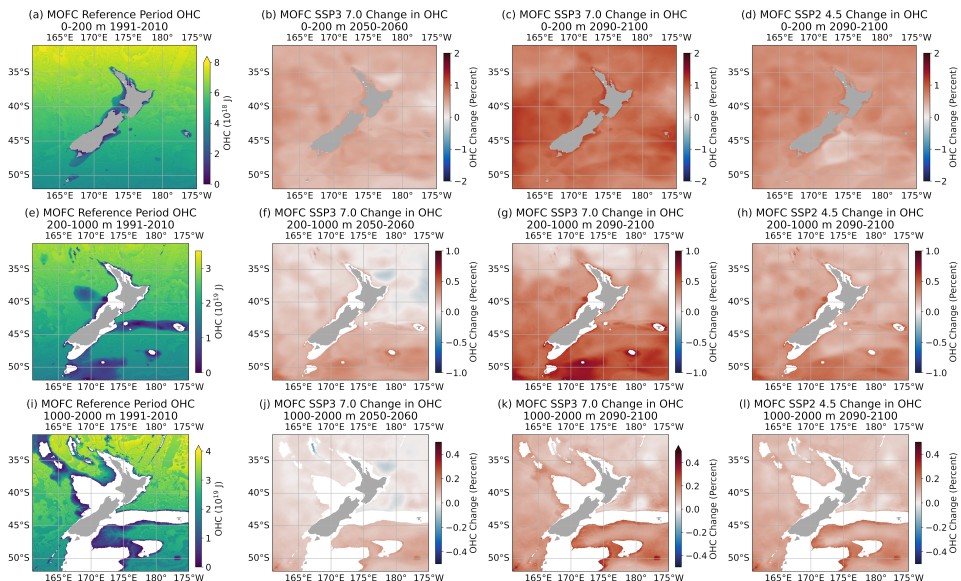

**Figure 8.** Reference period averaged Ocean Heat Content (in Joules) for the 0-200 m (a), 200-1000 m (e) and 1000-2000 m (i) layers. Percentage change between the average OHC in the last decade of each experiment and the reference period for SSP3 7.0 medium (b, f and j) and long-term (c, g and k) experiments, and SSP3 4.5 long-term experiment(d, h and l).

All these indicate changes in the ocean are dominated by upper ocean processes including local air-sea fluxes and advective heat transport. Some of this signal is captured in the ocean interior due to the central water-mass formation in the Subtropical Front region.

Indeed, the time-series of future climate OHC in Figure 5 shows how the upper ocean dominates the predicted changes. Compared to the MOFC reference period, the surface (0-200 m) ocean shows about 50% faster warming under both SSP3 7.0 experiments, while SSP2 4.5 shows moderately slower warming than observed during the reference period (Tables 1 and 3). In the 200-1000 m layer the warming is faster than the reference period MOFC and NZESM, but is comparable to the reference period MOH warming. The late-century SSP3 7.0 experiment displays the most rapid warming followed by SSP 2 4.5 with the mid-century SSP3 7.0 showing the slowest warming.

Between 1000 and 2000 m, warming during the 2030-2060 period is slower than during any of the reference period cases, while late-century warming in both SSP2 4.5 and SSP3 7.0 are comparable to reference period warming.





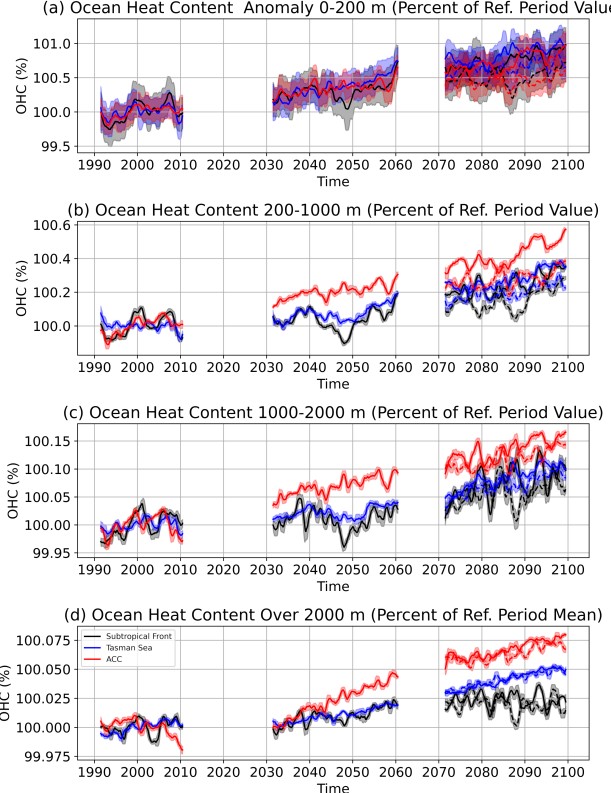

**Figure 9.** Time-series of Ocean Heat Content changes (as a reference period mean OHC) for the Tasman Sea (blue, 45-35S, <170E), Sub-Tropical Front (black, 40-45S, >174E) and ACC (red, <45S). Lines show the low-frequency component isolated using a 1 year rolling mean, solid lines correspond to the reference period or SSP3 7.0 and dashed lines to SSP2 4.5. Shaded areas indicate the standard deviation estimated in the same 1 year window used during averaging.

Figure 8 shows maps of the reference period OHC and its changes under future climate scenarios, as a percentage of the mean OHC of the reference period, for the depth layers of 0-200 m, 200-1000 m and 1000-2000 m depth layers. In the surface ocean (Figure 8a-d) there is warming over the entire domain, at its most intense in the Tasman Sea near 40-45°S and on the eastern margins of the Chatham Rise; the lowest warming occurs in the Tasman Front under both SSP3 7.0 experiments and offshore of the Southland Current in the SSP2 4.5 experiment. Between 200 m and 1000 m warming is strongest in the south

of the domain, with relatively slow warming in the Tasman Front (to the west of New Zealand) and north of the Chatham Rise (to the east); in the short term (2030-2060) experiment we even see slight cooling in near the East Cape Eddy. Below 1000 m warming is strongest on the continental slopes followed by the South East of the domain.

Based on the regional warming discussed above we examined regional changes in OHC in the Sub-tropical front east of New Zealand; the Tasman Sea south of the Tasman Front and the northern flank of the ACC. Times-series of OHC (as percentages

of the 1991-2010 mean value) for these regions are shown in Figure 9. We see that in the surface ocean all regions display



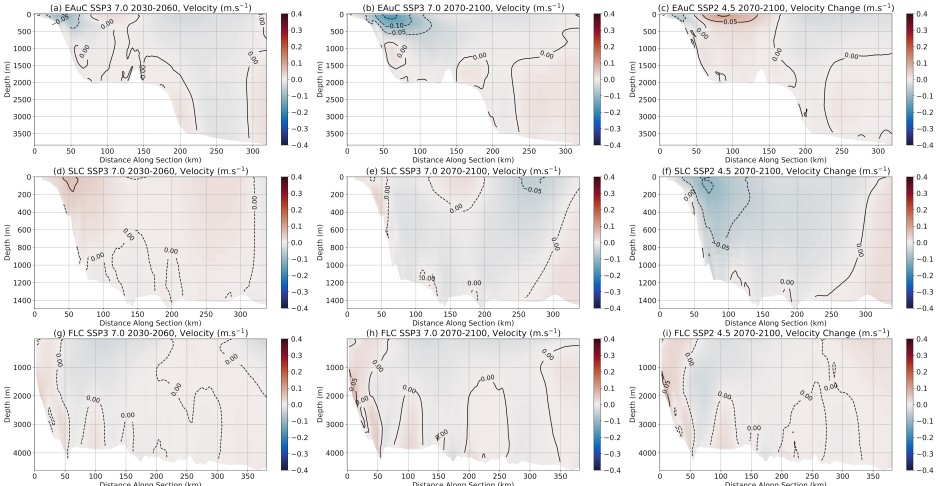

**Figure 10.** Changes in cross-section velocity relative to the reference period for the East Auckland Current (a-c); Southland Current (d-f) and Fiordland Current (g-i) under future climate scenarios.

similar levels of warming, likely controlled by surface forcing. At deeper levels we see the ACC warms much more rapidly than all other regions, presumably due to the higher level of connectivity (and, thus, younger "age") between Southern Ocean water masses and the atmosphere compared to elsewhere in the domain. We see that in the upper ocean the late century SSP2 4.5 generally shows less warming than SSP3 7.0, while in the deep ocean (1000 m and below) SSP2 4.5 and SSP3 7.0 show
similar levels of warming in all regions, likely due to the time-scales of connectivity to the surface introducing a significant time-lag.

## 5.2 Changes to Transport and Velocity

Cross-section velocities across the East Auckland Current; Southland Current and Fiordland Current for the three future climate scenarios are shown in Figure 10. Compared to the historical period (Figure 6b, f and j) large-scale circulation structures remain
robust. We see possible evidence of a weakening of the EAuC core under SSP3 7.0 and a 20-30 km offshore shift of the EAuC core under SSP2 4.5. The SLC and FLC show little evidence of change under all future climate.

We also examined the mean and standard deviation of transport across key currents under future climate experiments (Table 4). We find that transports through all sections show similar magnitudes and standard deviations as MOFC during the reference period. Transport power-spectra and mean seasonal cycles (not shown) also showed no significant changes from the reference
period.

These results suggest that the large-scale coastal circulation around New Zealand is only weakly sensitive to climate change.





**Table 4.** Mean Transport (Sv) and Standard Deviation on sections shown in Figure 1 under future climate scenarios.

| Section | SSP3 7.0 (Sv) 2030-2060 | SSP3 7.0 (Sv) 2070-2100 | SSP2 4.5 (Sv) 2070-1000 |
|---|---|---|---|
| (A) Fiordland Current | -6.60±4.04 | -7.78±4.73 | -7.72±4.54 |
| (B) Southland Current | 1.83±2.08 | 2.79±2.77 | 3.46±3.01 |
| (C) EAuC (North) | 13.93±7.88 | 13.02±6.64 | 17.35±8.16 |
| (D) EAuC (South) | 10.26±11.01 | 11.19±11.04 | 11.67±12.23 |
| (E) ECC | -29.18±10.95 | -34.88±11.67 | -36.35±13.31 |
| (F) West Coast of North Island | 5.36±2.49 | 5.39±2.81 | 5.29±2.89 |
| (G) Cook Strait | 0.33±0.27 | 0.35±0.27 | 0.45±0.33 |

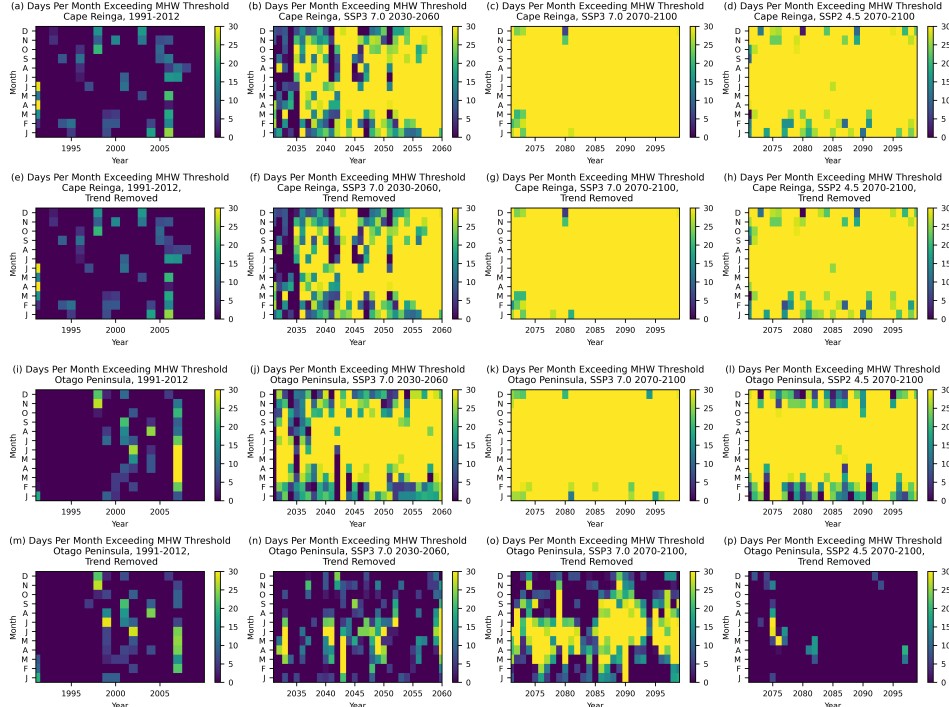

**Figure 11.** The number of days exceeding the MHW temperature binned by year and month in the Cape Reinga (a-h) and Otago Peninsula (i-p) regions regions with MHW threshold defined a fixed baseline (e-h and m-p) and a shifting baseline (a-d and i-l). Columns correspond to different experiments, from left to right: Reference period; SSP3 7.0 2030-2060; SSP3 7.0 2070-2100 and SSP2 4.5 2070-2100.

## 5.3 Marine Heat Waves

The occurrence of marine heat waves was investigated using a similar approach used to (Hobday et al., 2016), in which the reference period (1991-2010) MOFC SST data for each grid point was binned by day of the year, and for each day of year the

SST values corresponding to the 90% percentile were identified and used to define the marine heatwave threshold. There has been some discussion about whether the definition of MHWs should focus on a fixed baseline or a shifting baseline Chiswell





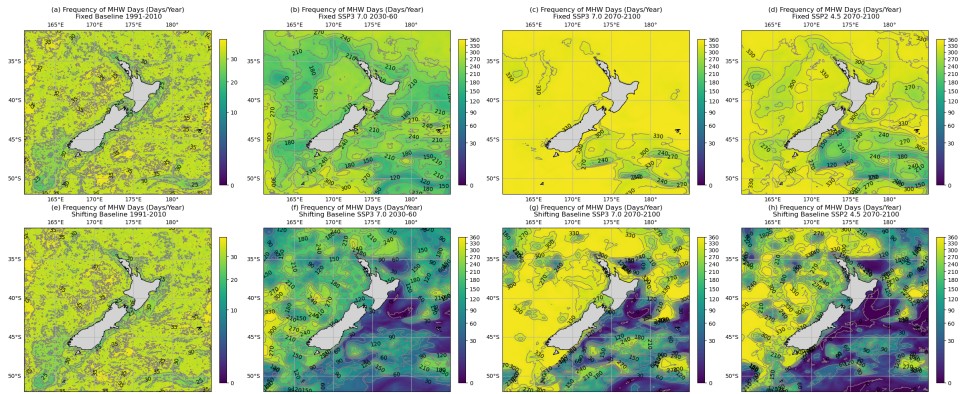

**Figure 12.** Frequency of MHW-days in days per year using thresholds calculated with a fixed baseline (top row) or with a shifting baseline for the reference period (a and e); SSP3 7.0 2030-2060 (b and f); SSP3 7.0 2070-2100 (c and g) and SSP2 4.5 2070-2100 (d and h).

(2022); Amaya et al. (2023), especially when considering future climate scenarios . To examine this we focused on the two regions indicated by green boxes in Figure 1: Cape Reinga (Box 1) and the Otago Peninsula (Box 2). Other regions were considered, but generally showed behaviour consistent with one of the two regions shown here. We computed time-series of

bin-averaged SST for the reference period. We then computed MHW thresholds, as above, for a fixed baseline (using reference period data without detrending) and for a shifting baseline (subtracting the reference period trend from the data both before defining threshold and from each future climate scenario). These MHW thresholds were then applied to the time series of the experiments and the number of days exceeding the MHW SST thresholds during the reference period were plotted as a function of year and month of year (Figure 11, a, e, i and m). We see that during the reference period there are few differences between

a fixed baseline and a moving baseline. The shifting baseline increases the number of MHW days early in the reference period and marginally decreases MHW days late in the reference period, but still preserves the overall distribution of MHW days as a function of year and month.

Under future climate scenarios the fixed baseline (Figure 11b-d and j-l) sees MHW-like conditions become much more frequent, with both regions showing year-round exceeding of the MHW threshold for both late-century scenarios. The medium-

term scenario (2030-2060) shows the onset of near-permanent MHW conditions throughout the mid-2040s to mid-2050s. Considering the occurrence of MHWs in the future climate under a shifting baseline (Figure 12f-h and n-p) we see little difference in the Cape Reinga region with MHWs occurring essentially at the same rate by both year and month as the non-detrended case examined above. However, in the Otago Peninsula region a shifting baseline drastically reduced the number of MHW days in all future climate scenarios. Under SSP3 7.0 2030-2060 with a shifting baseline, while MHWs are still more

frequent than the reference they remain irregular events. SSP3 7.0 2070-2100 shows MHW conditions dominating during winter months but being rare in summer, in contrast to the case fixed baseline where the region is under MHWs essentially all year. SSP2 4.5 2070-2100 shows an occurrence of extreme SST comparable to the 2030-2060 period.



**Table 5.** Average of percentage of model domain in each MHW category under each experiment. FB denotes Fixed Baseline and SB denotes Shifting Baseline

| MHW Category | Reference Period | | SSP3 7.0 2030-2060 | | SSP3 7.0 2070-2100 | | SSP2 4.5 2070-1000 | |
|---|---|---|---|---|---|---|---|---|
| | FB | SB | FB | SB | FB | SB | FB | SB |
| Moderate | 7.38 | 7.34 | 33.34 | 21.60 | 14.10 | 17.91 | 26.64 | 18.58 |
| Strong | 0.91 | 0.92 | 20.30 | 10.52 | 24.84 | 19.13 | 28.56 | 13.99 |
| Severe | 0.13 | 0.13 | 7.20 | 3.28 | 23.01 | 13.27 | 16.17 | 6.72 |
| Extreme | 0.03 | 0.04 | 3.16 | 1.36 | 30.71 | 13.41 | 10.53 | 4.31 |

We then expanded this analysis from regional boxes to a gridpoint-by-gridpoint view; resulting maps of the frequency of MHW for future climate experiments are shown in Figure 12. As expected, the reference period (Figure 12a and e) show essentially the same spatial pattern in the occurrence of MHW-days. We see that, using an MHW definition with a fixed basedline (Figure 12a-d), even over the 2030-2060 experiment MHWs occur significantly more than 10% of the time (36 days per year) over virtually the entire domain, with much of the New Zealand coast experiencing historically extreme SST more than 200 days per year. Moving into 2070-2100, SSP3 7.0 shows virtually the entire domain (excluding some isolated areas in the northern flank of the ACC) exceeding the Marine Heat Wave threshold over 330 days per year, putting essentially the entire model domain in permanent marine heat wave conditions. The situation under SSP2 4.5 is not quite as severe but still shows most New Zealand coastal waters under MHW conditions in excess of 270 days per year. Using a shifting baseline (Figure 12e-h) produces similar results to the above in the west and north of the domain, but shows a significantly lower frequency of days above the MHW threshold to the east and south of New Zealand, corresponding to the FLC-STF system. This region shows the strongest SST trend during the reference period, likely as a side effect of the incorrect placement of the STF east of New Zealand discussed previously.

The average percentages of the domain experiencing MHWs of particular categories for each experiment under a fixed baseline (FB) and shifting baseline (SB) are shown in Table 5. Moderate MHWs correspond to a SST anomaly of 1-2 times the threshold; Strong to 2-3 times the threshold; Severe to 3-4 times the threshold and Extreme to SST anomaly in excess of 4 times the threshold. As above, during the reference period the use of a shifting baseline makes no meaningful difference to the occurrence of any MHW category. Under SSP3 7.0 2030-2060 and SSP2 4.5 2070-2100 using a shifting baseline reduces the occurrence of Moderate MHWs by about a third and the occurrence of higher MHW categories by about half. Under SSP3 7.0 2070-2100 using a shifting baseline increases the percentage of the domain experiencing moderate MHWs by about 20% while decreasing the occurrence of strong MHWs by a third; Severe MHWs by a half and Extreme MHWs by a half.

The definition of a marine heat wave event should take into account the use one wishes for the analysis and carefully consider the political decision-making repercussions. For example, anyone interested in changes of current ecosystems and environmental impacts will want to compare future scenarios to a fixed baseline period or better, fixed threshold temperatures of known impact for species of particular interest. This is the framework of choice for any present decisions that will influence the next decades, as it properly takes into consideration the future impacts on our coastal ecosystems and the services they



provide. However, the definition of a new moving threshold emphasizes how what we understand today as an "extreme" will
change as our climate shifts and waters get overall warmer.

## 6   Conclusions

We have developed new marine climate dynamic downscaling simulations for the New Zealand region, combining the Moana
Ocean Hindcast (de Souza et al., 2022) model configuration with forcing from the New Zealand Earth System Model (Williams
et al., 2016; Behrens et al., 2020) and New Zealand Regional Climate Model. This dataset offers approximately 5km spatial
resolution and hourly temporal resolution for a 20 year historical baseline and three 30 year future climate experiments includ-
ing a mid-century high emissions scenario (SSP3 7.0); a late-century high emissions scenario (SSP3 7.0) and a late-century
medium emissions scenario (SSP2 4.5). These datasets provide better spatial and temporal resolution than previous regional
downscaling of future New Zealand marine climate, both improving representation of coastal and shelf processes and providing
data on scales relevant to the study of ecology and fisheries on the Continental Shelf.

We have evaluated our model against the Moana Ocean Hindcast (de Souza et al., 2022), a well-validated regional ocean
model that uses the same configuration that has been running for 30 years with historical data and has been deployed to
operations providing 7-day forecasts. We find that when considering SST; SSH; transport; temperature sections and velocity
sections, MOH and MOFC are in general agreement to the west and north of New Zealand. However, in the South-East of
the domain MOFC shows substantial differences to MOH: MOFC places the time-mean 12°C SST isotherm significantly
further south than MOH, with the isotherm approaching the New Zealand coast in proximity to the Otago Peninsula instead
of near the Banks Peninsula; the section across the Southland Current shows southward flow offshore of the SLC instead of
weak northward flow; and time-mean SSH shows a 'finger' of high SSH extending towards New Zealand from the eastern
boundary near 46 °S that is not present in MOH. These discrepancies are consistent with known issues with the placement of
the Subtropical Front in the version of the New Zealand Earth System Model used as boundary forcing in this study (Behrens
et al., 2020; Behrens). MOFC also produces higher estimates of ocean heat content than MOH, likely due to a combination of
the issues in the STF (discussed above) and the relatively strong climate sensitivity of NZESM (Behrens et al., 2022; Meehl
et al., 2020), which has transient and equilibrium climate sensitivities higher than the best current estimates of the likely range
(Nijsse et al., 2020). We conclude that, subject to known limitations of the NZESM data used to initialize and drive MOFC,
MOFC shows reasonable agreement with MOH.

We have used MOFC to explore future changes to water temperature; ocean heat content and coastal circulation. Consid-
ering T-S plots we see a shift towards warmer and fresher Subantarctic Mode Waters and Subtropical Mode Waters, while
intermediate and deep waters show few changes. Changes in temperature sections on major currents are predominately surface
intensified, and this is reflected in maps and time-series of OHC. We observe that the Tasman Sea, STF east of New Zealand
and the ACC regions display similar rates of warming in the upper ocean under all future climate scenarios. However, at deeper
levels, the ACC shows more rapid warming than the other regions, and in the deep ocean by the late century (both SSP3 7.0 and
SSP2 4.5 scenarios) the Tasman Sea shows more rapid warming than the STF. We find little change in the time-mean transport



or time-mean structure of three major coastal ocean currents around New Zealand under all three future climate scenarios similar to Sen Gupta et al. (2021).

We have examined the frequency of Marine Heat Wave events using the definition given in Hobday et al. (2016), under both
historical and future climate conditions, and explored the effects of using historical time-mean and shifting (i.e. detrending the data before analysis) baselines. We find that during the historical period the different baselines make little significant difference to the occurrence of extremely high SSTs. However, under future climate scenarios time-mean and shifting MHW thresholds can result in significant differences. We see a strongly reduced occurrence of time-averaged days per year over the MHW threshold around much of the East of New Zealand corresponding to the areas showing the most rapid increase in SST, and in
that region a delay in onset of near year-round MHW conditions from the 2040s to the 2070s. We suggest that for historical to near future studies a fixed MHW baseline is appropriate, but more sophisticated approaches, such as MHW categories, are necessary for examining MHW occurrence further into the future.

This new dataset will be used for analyses including Lagrangian particle tracking studies to examine the effects of climate change on larval dispersal of major commercial fisheries including Green Lipped Mussels; Abalone and Scampi.

*Code and data availability.*

The Regional Ocean Modelling System is available from https://www.myroms.org. This study employed ROMS v4.0. The Moana Future Climate ROMS configuration is available on Zenodo from https://doi.org/10.5281/zenodo.13901412 (Roach, 2024a).

Moana Ocean Future Climate hourly and daily average temperature salinity and velocity fields are available from https:
//www.moanaproject.org/data-webform. Sample 3D daily fields are available on Zenodo from https://doi.org/10.5281/zenodo. 10811116 (Roach, 2024b).

Moana Ocean Hindcast daily average temperature, salinity and velocity fields are available from https://www.moanaproject. org/data-webform, sample fields are also available on Zenodo from https://zenodo.org/record/5895265 (de Souza, 2022).

NZESM historical sample fields can be accessed at http://doi.org/10.5281/zenodo.3581390. For access to the full dataset
contact NIWA via https://niwa.co.nz/contact.

*Author contributions.*

CR developed, ran and analysed the Moana Ocean Future Climate experiments. JS developed the Moana Ocean Hindcast configuration from which MOFC is derived and provided useful guidance on the analysis of MOFC. EB and SS provided access to and pre-processing of the NZESM and NZRCM data used as boundary and surface forcing.
All four authors contributed to the writing of this paper.



*Competing interests.*

The contact author has declared that none of the authors has any competing interests.

*Acknowledgements.* This work is a contribution to the Moana Project (www.moanaproject.org), funded by the New Zealand Ministry for Business Innovation and Employment, contract number METO1801.

We wish to thank NIWA for providing the New Zealand Earth System Model future climate output used in this study.

Development of NZESM was supported by funding from the Ministry for Business Innovation and Employment via C01X1902 Deep South National Science Challenge.

The authors wish to acknowledge the contribution of NeSI (http://www.nesi.org.nz) to the results of this research. New Zealand's national compute and analytics services and team are supported by the New Zealand eScience Infrastructure (NeSI) and funded jointly by NeSI's

collaborator institutions and through the Ministry of Business, Innovation and Employment.



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
