# Peer review of "Moana Ocean Future Climate V1.0: High Resolution Marine Climate Futures For The New Zealand Region"

_EGUsphere, 2024_

## Referee Comment (RC1)

The article titled "Moana Ocean Future Climate V1.0: High Resolution Marine Climate Futures For The New Zealand Region" by Roach et al., outlines a historical and future downscaling simulations of the Earth System Model configuration known as NZESM using the ROMS regional ocean model in combination with a downscaling of the NZESM atmosphere. The manuscript provides a validation of the model against a downscaled reanalysis and then proceeds to consider and document future changes. As I am not aware of a publication documenting the results of the NZESM future simulations outside of marine heat waves this downscaling analysis is particularly valuable for those wanting to understand possible change in the ocean around New Zealand. I recommend the article be accepted pending some major revisions and improvements to readability that I now discuss.

It is a bit unclear to me what the primary objectives are in documenting the new modelling framework within this manuscript. Where does the originality lie in this setup that the authors wish to highlight? It would be good to have some greater clarity about this. Considering future changes in the ocean around New Zealand is certainly achieved. But in a couple of places, the authors are clear that "the behaviour of the ocean current systems around New Zealand is sensitive to mesoscale, sub-mesoscale and high-frequency dynamics such as eddies, tides and transient responses to wind" implying that the results from the downscaling will be better than the driving NZESM model. However, there is no attempt to show this is true. The NZESM simulations are at 15km and the ROMS simulations are at 5km. Has this increased resolution improved anything? The results comparing historical simulations simply suggest that ROMS inherits its biases in temperature and salinity from NZESM. Furthermore, a key component of the modelling setup is the use of a high-resolution downscaling of the NZESM atmosphere to 12km, known as NZRCM. This then provides atmospheric forcing to ROMS at much higher resolution. This might be expected to be useful, but is this expensive computational exercise worth it? The analysis makes no attempt to answer this question as far as I can tell. An experiment that used exclusively NZESM atmospheric forcing over the domain rather than NZRCM would allow for a direct comparison and address the question of whether a high-resolution atmosphere has added value. Regardless I believe it is at a minimum necessary to demonstrate how this setup is more useful, or somehow different than NZESM, and including results from NZESM in the present figures and any appropriate new ones should allow for a straightforward comparison. Please see the attached pdf file for additional line-by-line comments.

Specific comments:

L2 "CMIP6 reference conditions; SSP2-4.5 and SSP3-7.0 emissions trajectories". The IPCC reports do not ever refer to CMIP6 reference conditions, and it is unclear what exactly is being referred to, while SSP2-4.5 and 3-7.0 are two illustrative emissions and concentrations scenarios, which the IPCC does not describe as trajectories. Change throughout manuscript.

L3 Unclear what ROMS is and hence unclear that the forcings being referred to are 15km output from a global NEMO ocean model, a component of the Earth System Model known as NZESM.

L6 Unclear what a Moana Ocean Hindcast is.

L9 Ocean Heat Content should not be capitalized and there is an implication that a mode water is something different from a component of the upper ocean.

L11 East should not be capitalized

L11 Marine Heat Waves (MHWs) should be marine heatwaves (MWHs) here and throughout the manuscript.

L11 The key finding appears to relate to highlighting a known methodological question about marine heatwaves which is not surprising,  rather than summarising the actual key changes seen in the ocean around New Zealand.

L15 Usually written "Aotearoa New Zealand's"

L15 marine domain presumably refers to exclusive economic area

L18 reference required for claim about ecosystem services

L51 I'm a bit confused. There is no ecosystem analysis in this paper. The analysis focuses on temperature, including increases in marine heat wave events because of their known importance to marine ecosystems around NZ. Which requires references.

L63 Worth defining what is meant by downscaling. I for one do not consider the NZ Earth System Model to be a regional downscaling as claimed at line 64, while I consider that NZRCM and MOH are.

L72 It would be useful to briefly explain that MOH is a downscaling of a historical ocean reanalysis. Moreover, the authors should simply reference GMD - Moana Ocean Hindcast – a > 25-year simulation for New Zealand waters using the Regional Ocean Modeling System (ROMS) v3.9 model which is the appropriate reference for MOH.

L100 It would be useful to explain earlier that NZESM has a two-way nesting to 12-20km resolution in the New Zealand region – possibly why the authors refer to it earlier as a downscaling.

L106 Details of the historical NZESM simulation should be noted. Important to acknowledge the NZESM historical experiment used here is not nudged or some form on reanalysis,  but a free running historical simulation with prescribed external forcings. This is noted later in the manuscript.

L113 higher end of warming relative to the full suite of CMIP6 ocean model under each of the SSPs.

L145 It is elsewhere outlined that there is an ensemble of three historical NZESM simulations available. Presumably only on of these has been used here. And when the results are compared with NZESM it is with the one ensemble member used for the MOH experiment.

Figure 2. Would be good to include labelling of mode waters on this figure as authors uses modes to discuss these changes but readers may not be familiar with local water modes.

Figure 3. Fonts are too small on figures here and elsewhere with multiple panels

L175 As explained earlier I think one of the primary purpose of this paper should be to show that the new downscaling is an improvement over NZESM, or adds value in some way. As such why not just include NZESM in figure 3, as was down for figure 2, rather than referring vaguely to another manuscript. At this point the summary would be that errors in SST and salinity are very similar in MOHC to NZESM.

Table 1 should ideally appear after heading 4.2

L207 Again it should be possible to show in Figure 5 that the bias is coming from NZESM, and possibly due to its high climate sensitivity.

L204. I'm not sure what is meant by seasonal variability. Figure 5 caption suggests the shading is the standard deviation of the values uses to calculate the mean. In order to qualitatively consider whether the means are different it may be more useful to show a formal confidence interval.

L287 How do these results compare to Behrens, E., Rickard, G., Rosier, S., Williams, J., Morgenstern, O., and Stone, D.: Projections of Future Marine Heatwaves for the Oceans425 Around New Zealand Using New Zealand's Earth System Model, Frontiers in Climate, 4, https://doi.org/10.3389/fclim.2022.798287,2022. Has increased resolution provided additional information?

L349 "both improving representation of coastal and shelf processes and providing data on scales relevant to the study of ecology and fisheries on the Continental Shelf". While I imagine this might be true I did not see evidence for this presented in the manuscript.

L380 "We suggest that for historical to near future studies a fixed MHW baseline is appropriate, but more sophisticated approaches, such as MHW categories, are necessary for examining MHW occurrence further into the future." Such a definitive conclusion is perhaps unwarranted. As noted earlier in the manuscript the appropriate method depends on the question being asked regarding marine heatwaves, rather than simply on the timeframe.